# Primary Cutaneous CD8+ Aggressive Epidermotropic Cytotoxic T-Cell Lymphoma: A Rare and Aggressive Case Report with Clinical and Pathological Insights

**DOI:** 10.3390/jcm14217842

**Published:** 2025-11-05

**Authors:** Janyna Jaramillo, Katty Méndez-Flores, Nataly Lascano, Santiago Palacios-Álvarez, Marlon Arias-Intriago, Juan S. Izquierdo-Condoy

**Affiliations:** 1CEPI Centro de la Piel, Quito 170508, Ecuador; 2Dermatology Department, Hospital de Especialidades Carlos Andrade Marin, Quito 170103, Ecuador; 3Pathology Department, SOLCA Quito, Quito 170138, Ecuador; 4One Health Research Group, Universidad de las Américas, Quito 170124, Ecuador

**Keywords:** cutaneous T-cell lymphoma, primary cutaneous CD8+ T-cell lymphoma, JAK-STAT pathway

## Abstract

**Introduction:** Primary cutaneous CD8+ aggressive epidermotropic cytotoxic T-cell lymphoma (PCAETL) is a rare and highly aggressive subtype of cutaneous T-cell lymphoma (CTCL), accounting for less than 1% of CTCL cases. It is defined by CD8+ cytotoxic T-cell proliferation with marked epidermotropism, necrosis, and a high proliferative index. Clinically, it presents as ulcerated or necrotic lesions with rapid progression and poor response to conventional therapies. **Aims:** To describe a fatal case of PCAETL in a young adult female, emphasizing the diagnostic challenges, clinical progression, histopathological features, and treatment limitations. **Case Presentation:** A 41-year-old Venezuelan woman presented with a 10-month history of disseminated papules and nodules initially misdiagnosed as Hansen’s disease. After her arrival in Ecuador, she was re-evaluated and found to have generalized dermatosis with ulcerated nodules and tumors. Histopathological examination revealed atypical epidermotropic CD8+ T-cell infiltration with extensive necrosis. Immunohistochemistry demonstrated strong positivity for CD3, CD5, and CD8, and a Ki-67 index of 80%, confirming the diagnosis of PCAETL. The patient received methotrexate with partial response but experienced disease relapse during second-line etoposide therapy. She developed febrile neutropenia and died five months after diagnosis. **Conclusions:** This case highlights the rarity, diagnostic complexity, and aggressive nature of PCAETL. Early recognition and clinico-pathological correlation are essential for timely diagnosis. However, therapeutic options remain limited, and outcomes are poor despite systemic chemotherapy. Further research into targeted and personalized therapies is urgently needed to improve survival in this devastating disease.

## 1. Introduction

Primary cutaneous CD8+ aggressive epidermotropic cytotoxic T-cell lymphoma (PCAETL) is a rare and highly aggressive subtype of cutaneous T-cell lymphoma (CTCL), accounting for less than 1% of all CTCL cases. It is characterized by the neoplastic proliferation of CD8+ cytotoxic T lymphocytes with marked epidermotropism, epidermal necrosis, and a high proliferative index. Clinically, it typically presents with localized or generalized skin lesions, often involving acral regions, and may also affect mucosal surfaces. Histologically, PCAETL is defined by the infiltration of atypical CD8+ lymphocytes into the epidermis, resulting in ulceration and extensive necrosis. The skin manifestations may include well-demarcated plaques, tumors, or diffuse papules and nodules, frequently displaying a centralperipheral pattern of erosion or necrosis [1,2,3].

PCAETL primarily affects adults, with a reported male-to-female ratio of approximately 3:1, and most cases occur between the ages of 40 and 60. In contrast to mycosis fungoides (MF)—which typically follows an indolent course with patch, plaque, and tumor stages—PCAETL exhibits a rapid clinical progression and shows limited response to systemic chemotherapy, often resulting in a poor prognosis. The cytotoxic CD8+ phenotype of this lymphoma was first described by Agnarsson in 1990, and the term “aggressive epidermotropic CD8+ T-cell lymphoma” was introduced by Berti et al. in 1999 [2,4].

Here, we report the case of a 41-year-old female diagnosed with PCAETL, emphasizing the disease’s rapid and aggressive progression. Although an initial transient response to chemotherapy was observed, the disease advanced during treatment.

## 2. Case Presentation

A 41-year-old Venezuelan woman with no relevant medical history developed progressive skin lesions over 10 months, initially misdiagnosed as Hansen’s disease and treated without improvement. After relocating to Ecuador, she presented with generalized dermatosis involving >5% of body surface, showing papules, nodules, and ulcerated tumors up to 8 cm, with crusting, plaques, and foul-smelling discharge. Sensory function remained intact, and hands, feet, and genitalia were spared (Figure 1).

Because of discrepancies between the clinical findings and the initial diagnosis, a biopsy of a right arm nodule was performed along with additional studies. Laboratory tests, including complete blood count, electrolytes, liver and kidney function, and uric acid, were normal. However, LDH was elevated at 351 U/L and β2-microglobulin at 2.9 mg/L. Serologies for HBV, HCV, HTLV-1, and HIV were negative, as were fungal cultures. Chest CT demonstrated multiple, well-defined nodular lesions throughout both lungs, more prominent in the right lower lobe.

Histopathological analysis of the skin biopsy showed intense proliferation of atypical lymphoid cells extending into the subcutaneous tissue, with marked epidermotropism. The neoplastic cells exhibited moderate pleomorphism, hyperchromatic and irregular nuclei of medium and small size, and atypical mitotic figures (Figure 2).

Immunohistochemistry revealed strong membranous positivity for CD3 (Figure 2), CD5, and CD8 in the neoplastic cells, and was negative for CD4, CD7, CD10, CD20, CD56, and granzyme B. The proliferation index, as assessed by Ki-67, was approximately 80%. Based on these findings, a diagnosis of primary cutaneous aggressive epidermotropic CD8+ cytotoxic T-cell lymphoma (PCAETL) was established.

The patient was referred to the Clinical Oncology service in September 2022 and initiated on first-line treatment with methotrexate. After 10 cycles, a partial clinical response was observed. She was subsequently switched to second-line chemotherapy with etoposide. However, after three cycles, the patient developed high-risk febrile neutropenia requiring intensive care unit (ICU) admission, protective isolation, and intravenous antibiotics. Within two weeks, new papular lesions emerged, some of which became ulcerated and disseminated rapidly, accompanied by the appearance of generalized lymphadenopathy, consistent with disease relapse. The patient died in January 2023, five months after her diagnosis.

## 3. Discussion

Primary cutaneous aggressive epidermotropic CD8+ cytotoxic T-cell lymphoma is the most aggressive subtype of CD8+ cutaneous T-cell lymphomas (CTCLs). First described by Berti et al. in 1999, PCAETL is characterized by rapid progression and poor clinical outcomes [4]. Clinically, it presents with ulcerated, often necrotic plaques or tumors that may involve mucosal surfaces and acral regions such as the palms and soles [5,6]. Although lymph node involvement is uncommon, extracutaneous dissemination to the lungs, testicles, central nervous system, and oral mucosa has been reported. Localized disease frequently evolves into disseminated involvement, and in advanced stages, may mimic pyoderma gangrenosum. Unlike other aggressive lymphomas, PCAETL is not typically associated with immunosuppression [1].

PCAETL is exceedingly rare, accounting for less than 1% of all T-cell lymphomas, and occurs more frequently in males, with a mean age of onset around 77 years. Despite its low incidence, the prognosis is poor, with a five-year overall survival rate of approximately 18% and a median survival of 23–32 months [1].

CTCLs represent a heterogeneous group of non-Hodgkin lymphomas that primarily affect the skin [1,3]. Most CTCLs exhibit a CD4+ helper T-cell phenotype, but CD8+ variants form a distinct subset with diverse clinical behaviors and prognoses [7]. The spectrum of CD8+ CTCLs ranges from indolent forms—such as CD8+ MF and lymphomatoid papulosis (LyP) type D—to aggressive entities like PCAETL. CD8+ MF, while rare (less than 5% of MF cases), is the most common CD8+ CTCL subtype. LyP type D is marked by self-resolving papulonodular lesions with an epidermotropic CD8+ infiltrate and CD30 positivity. SPTCL typically presents with deep subcutaneous tumors or plaques mimicking panniculitis, showing a CD4^−^/CD5^−^/CD8+ phenotype and expression of cytotoxic markers such as granzyme B and TIA-1. PCGDTL, which may not consistently express CD8, is another aggressive subtype, often presenting with erythematous to violaceous plaques or nodules with ulceration and necrosis [7].

Recent studies suggest that T-cell receptor (TCR) gene rearrangements and JAK2 gene fusions or activating mutations in the Janus kinase/signal transducer and activator of transcription (JAK/STAT) pathway may aid in the diagnosis of PCAETL [8].

The diagnosis of CD8+ CTCLs requires a comprehensive integration of clinical findings, histopathology, immunohistochemistry (IHC), and molecular studies. Histological features typically include a dense epidermotropic CD8+ lymphoid infiltrate with hyperchromatic nuclei and, in aggressive variants, extensive necrosis. However, CD8 positivity alone is not sufficient for diagnosis, nor is it prognostically decisive [7].

In the differential diagnosis of ulcerated or necrotic dermatoses, PCAETL may mimic several malignant and inflammatory conditions such as cutaneous angiosarcoma, pyoderma gangrenosum, and ulcerated cutaneous metastases. Cutaneous angiosarcoma is a rare but highly aggressive vascular tumor, most frequently involving the scalp and face of elderly individuals. It typically presents as multifocal violaceous or bruise-like plaques that may ulcerate or bleed. Histopathologically, it shows atypical endothelial proliferation with expression of vascular markers CD31, CD34, and factor VIII-related antigen, which help distinguish it from lymphoid neoplasms [9]. Pyoderma gangrenosum, in contrast, is a neutrophilic dermatosis characterized by painful ulcers with undermined violaceous borders and a rapidly progressive course. Biopsy reveals dense sterile neutrophilic infiltration without malignant cells, confirming its reactive rather than neoplastic nature [10,11]. Ulcerated cutaneous metastases, which occur in 0.7–10% of patients with advanced internal malignancies, often present as firm nodules or plaques that can ulcerate. The most common primary sites include breast, lung, and colorectal carcinomas. Histologically, they show glandular or epithelial morphology with cytokeratin and organ-specific marker positivity (e.g., TTF-1, CDX2), which aids differentiation from cutaneous lymphomas [12,13].

A wide differential diagnosis must be considered when evaluating CD8+ CTCLs. Table 1 outlines the key distinguishing features of PCAETL and other related disorders.

The pathogenesis of PCAETL is closely linked to dysregulation of the JAK/STAT signaling pathway. In most cases, this is driven by JAK2 gene fusions, which are unique to CD8+ PCAETL and not observed in other cytotoxic CTCLs. In fusion-negative cases, gain-of-function mutations in JAK2, STAT3, or STAT5B, or loss of negative regulators such as SH2B3, have been implicated. These alterations promote uncontrolled proliferation and survival of malignant T-cells, contributing to the aggressive clinical course [8].

Histologically, PCAETL shows a dense monomorphic to pleomorphic infiltrate of CD8+ T-cells with hyperchromatic, often indented nuclei and scant cytoplasm. Epidermal necrosis is a hallmark, and in advanced lesions, necrosis may extend into the dermis. Pagetoid epidermotropism is frequently observed, and spongiosis may result in blister formation. Tumor-stage lesions often exhibit diffuse or patchy dermal involvement, and “rimming” of subcutaneous fat lobules may be seen [1,4].

The immunophenotype is typically CD3+, CD8+, CD4^−^. However, CD8 expression may be weak or occasionally absent [22]. Tumor cells usually express βF1 (TCR-β), cytotoxic proteins such as TIA-1, granzyme B, and perforin, and exhibit a high Ki-67 proliferation index (>75%). Some cases display CD45RA positivity and lack CD2 and CD5, indicating a naïve T-cell phenotype. TCR-delta and Epstein–Barr virus-encoded RNA (EBER) are negative. CD7 and CD56 are variably expressed, and most cases are CCR4-negative. IHC can also be used to demonstrate JAK/STAT activation via phosphorylated STAT3 (pSTAT3) and STAT5 (pSTAT5) [8,23]. These findings, together with clinical and histologic data, are essential for accurate diagnosis.

Treatment of PCAETL is particularly challenging due to its rarity, aggressive nature, and lack of clinical trial data. Patients are frequently excluded from trials because of rapid disease progression. Initial management often involves multi-agent chemotherapy and/or localized radiation therapy; however, responses are usually short-lived. Interferon-alpha has been reported to worsen the disease and is therefore contraindicated [1]. he median survival is approximately 12 months, and no significant prognostic differences have been found between cases with small versus large cell morphology or localized versus disseminated disease [20].

Early diagnosis and intervention are critical. In selected patients who respond to chemotherapy, allogeneic hematopoietic stem cell transplantation may offer a potential survival benefit. Moreover, next-generation sequencing can help identify actionable mutations, enabling personalized therapeutic strategies. Nevertheless, the overall prognosis remains poor, emphasizing the urgent need for further research into the molecular pathogenesis and treatment of PCAET.

## 4. Conclusions

Primary cutaneous CD8+ aggressive epidermotropic cytotoxic T-cell lymphoma represents one of the most challenging and fatal entities within the spectrum of cutaneous T-cell lymphomas. This case underscores the diagnostic complexity and fulminant course of the disease, as well as the need for a high index of suspicion when evaluating ulcerated or necrotic dermatoses unresponsive to conventional therapy. Histopathological and immunophenotypic confirmation are essential to differentiate PCAETL from other CD8+ lymphoproliferative disorders and inflammatory mimickers. Despite transient responses to chemotherapy, the prognosis remains dismal, reflecting the limited efficacy of current therapeutic options. Advances in molecular profiling, particularly regarding JAK/STAT pathway alterations, offer promise for the development of targeted therapies. Early multidisciplinary recognition and the integration of molecular diagnostics into routine practice are crucial to improve diagnostic accuracy and potentially guide future personalized treatment approaches for this rare and aggressive lymphoma.

## Figures and Tables

**Figure 1 jcm-14-07842-f001:**
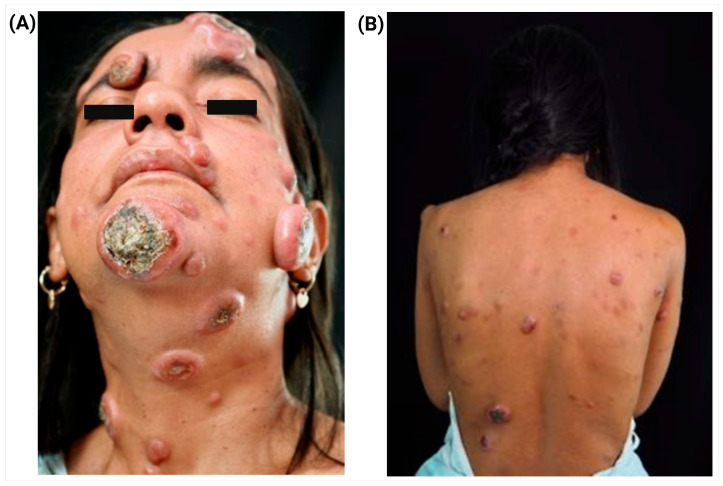
Clinical presentation of PCAETL. (**A**) Lesions involving the face and neck, presenting as erythematous-violaceous infiltrated nodules and tumors, some with central ulceration and superficial crusting. (**B**) Lesions on the back, characterized by diffuse erythematous-violaceous papules and fixed nodules without ulceration. Overall, the lesions range from 0.5 to 8 cm in diameter and involve more than 5% of the body surface area. The hands, feet, and genital regions are spared.

**Figure 2 jcm-14-07842-f002:**
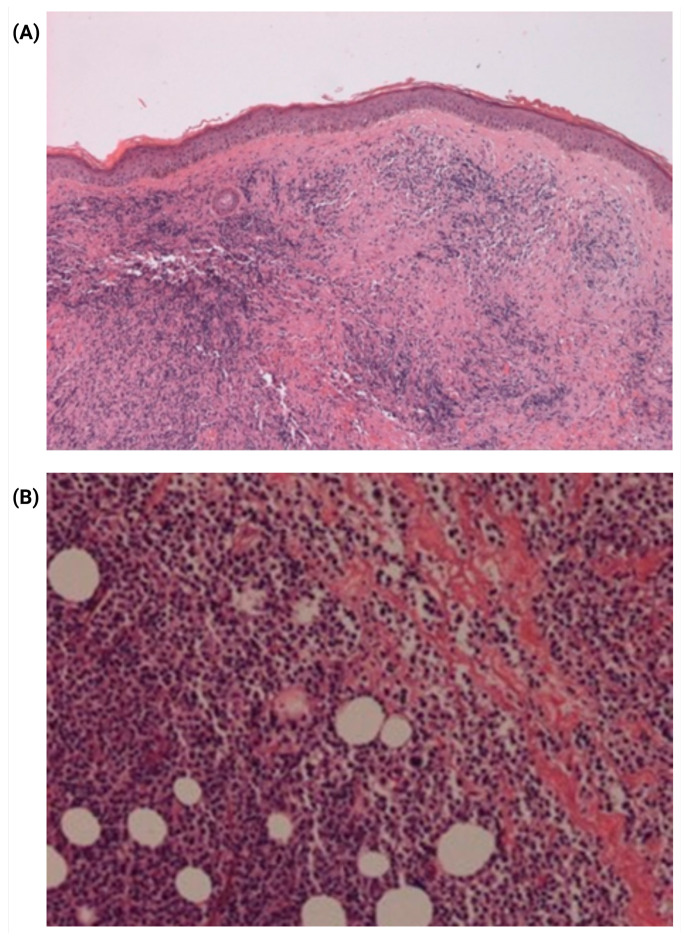
Histopathological features of skin biopsy. (**A**) Hematoxylin and Eosin (H&E) stain at 40× magnification showing dense infiltration of atypical lymphoid cells extending into the subcutaneous tissue with marked epidermotropism. (**B**) H&E stain at 400× magnification revealing lymphocytes with moderate pleomorphism, medium to small hyperchromatic and irregular nuclei, and atypical mitotic figures.

**Table 1 jcm-14-07842-t001:** Differential diagnosis of Primary Cutaneous CD8+ Aggressive Epidermotropic Cytotoxic T-Cell Lymphoma.

Entity	Key Features	Clinical Behavior	Distinguishing Features	References
CD8+ Mycosis Fungoides (MF)	Rare CD8+ variant; presents with patches, plaques, or tumors.	Indolent course	Less aggressive than other CD8+ CTCLs; histopathology and clinical behavior differentiate.	[2,7,14,15].
Lymphomatoid Papulosis (LyP), Type D	Self-resolving papulonodular lesions; epidermotropic CD8+ infiltrate, CD30+.	Benign course	Rarely progresses to lymphoma; CD30 positivity.	[2,7,14,16].
Subcutaneous Panniculitis-like T-Cell Lymphoma (SPTCL)	Deep subcutaneous tumors/plaques; CD8+, cytotoxic markers+.	Variable, often indolent	Lacks epidermotropism; mimics panniculitis.	[2,7,14,17].
Primary Cutaneous Gamma/Delta T-Cell Lymphoma (PCGDTL)	Highly aggressive; may mimic CD8+ AECTCL.	Aggressive	Expresses gamma/delta T-cell receptors (not alpha/beta).	[2,7,14,18].
Acral CD8+ T-Cell Lymphoproliferative Disorder	Solitary nodules on face, nose, or ears.	Indolent, favorable prognosis	Benign course; no systemic involvement.	[2,7,14,19].
Peripheral T-Cell Lymphoma Not Otherwise Specified (PTCL-NOS)	Cutaneous involvement possible; diagnosis of exclusion.	Variable	Lacks specific features of CD8+ CTCLs.	[2,7,14,20].
Natural Killer T-Cell Lymphoproliferative Disorders	Includes ENKTL (nasopharynx nodules) and HVLLPD (photo-exposed papulovesicular lesions).	Aggressive (ENKTL); variable (HVLLPD)	Distinct immunophenotype and clinical presentation.	[2,7,14,21].

## Data Availability

Data sharing is not applicable to this article as no datasets were generated or analyzed during the current study.

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
