# Peer review of "Primary Cutaneous CD8+ Aggressive Epidermotropic Cytotoxic T-Cell Lymphoma: A Rare and Aggressive Case Report with Clinical and Pathological Insights"

_jcm, 2025, doi:10.3390/jcm14217842_

Round 1

Reviewer 1 Report

Comments and Suggestions for Authors

The submitted case report presents an uncommon and clinically significant entity — Primary Cutaneous CD8+ Aggressive Epidermotropic Cytotoxic T-Cell Lymphoma (PCAETL). The authors describe a challenging and instructive case, emphasizing the diagnostic uncertainty that initially led to a misinterpretation and the importance of multidisciplinary evaluation in achieving the correct diagnosis.

            The manuscript is clearly written, well organized, and the histopathological and immunohistochemical descriptions are detailed. The case has educational clear value. For surgeons and dermatologists who often encounter such lesions, this report underscores the essential role of timely biopsy and thorough histological assessment before considering any reconstructive procedure. The discussion is concise and relevant, though it could be slightly expanded to include a few sentences on the differential diagnosis with other ulcerated malignancies such as angiosarcoma, pyoderma gangrenosum, or ulcerated metastases.

Finally, while the authors indicate that patient consent was obtained, a clear statement regarding ethical approval of institution where she was cured to publish medical data of patient would be necessary.

In summary, this is a well-prepared and informative manuscript on a rare and aggressive cutaneous lymphoma. It provides both clinical and pathological insights, is educational.

Author Response

Point by point letter

Re: “Primary Cutaneous CD8+ Aggressive Epidermotropic Cytotoxic T-Cell Lymphoma: A Rare and Aggressive Case Report with Clinical and Pathological Insights”

Reviewer 1

-The submitted case report presents an uncommon and clinically significant entity — Primary Cutaneous CD8+ Aggressive Epidermotropic Cytotoxic T-Cell Lymphoma (PCAETL). The authors describe a challenging and instructive case, emphasizing the diagnostic uncertainty that initially led to a misinterpretation and the importance of multidisciplinary evaluation in achieving the correct diagnosis.

We appreciate the reviewer’s thoughtful summary and recognition of the manuscript’s clinical and educational value.

-The manuscript is clearly written, well organized, and the histopathological and immunohistochemical descriptions are detailed. The case has educational clear value. For surgeons and dermatologists who often encounter such lesions, this report underscores the essential role of timely biopsy and thorough histological assessment before considering any reconstructive procedure. The discussion is concise and relevant, though it could be slightly expanded to include a few sentences on the differential diagnosis with other ulcerated malignancies such as angiosarcoma, pyoderma gangrenosum, or ulcerated metastases.

We appreciate this valuable suggestion. The Discussion section has been expanded to include a detailed paragraph addressing the differential diagnosis with other ulcerated or necrotic malignancies, as recommended.

-Finally, while the authors indicate that patient consent was obtained, a clear statement regarding ethical approval of institution where she was cured to publish medical data of patient would be necessary.

Thanks for your comment. We have added an explicit statement to the Ethics section of the manuscript.

-In summary, this is a well-prepared and informative manuscript on a rare and aggressive cutaneous lymphoma. It provides both clinical and pathological insights, is educational.

We appreciate the reviewer’s kind remarks and have carefully reviewed the manuscript for clarity and conciseness.

Reviewer 2 Report

Comments and Suggestions for Authors

The presented case represents a valuable contribution to the existing literature. It concerns an exceptionally rare form of lymphoma, which poses substantial diagnostic challenges and often leads to diagnostic delay, disease progression, and ultimately, poor clinical outcomes. I believe that this reports is of particular importance, as they can raise clinicians’ awareness and prompt consideration of this diagnosis when encountering unusual cutaneous lesions. I congratulate the authors on their meticulous presentation of this remarkable and instructive case.

Author Response

Point by point letter

Re: “Primary Cutaneous CD8+ Aggressive Epidermotropic Cytotoxic T-Cell Lymphoma: A Rare and Aggressive Case Report with Clinical and Pathological Insights”

Reviewer 2

The presented case represents a valuable contribution to the existing literature. It concerns an exceptionally rare form of lymphoma, which poses substantial diagnostic challenges and often leads to diagnostic delay, disease progression, and ultimately, poor clinical outcomes. I believe that this reports is of particular importance, as they can raise clinicians’ awareness and prompt consideration of this diagnosis when encountering unusual cutaneous lesions. I congratulate the authors on their meticulous presentation of this remarkable and instructive case.

We thank the reviewer for their positive evaluation. Although no specific revisions were requested, minor stylistic and grammatical refinements were made throughout the text to enhance clarity and readability.